# Parting with Misconceptions about Learning-based Vehicle Motion Planning

**Daniel Dauner**[1,2]     **Marcel Hallgarten**[1,3]     **Andreas Geiger**[1,2]     **Kashyap Chitta**[1,2]
[1]University of Tübingen     [2]Tübingen AI Center     [3]Robert Bosch GmbH
https://github.com/autonomousvision/tuplan_garage

**Abstract:** The release of nuPlan marks a new era in vehicle motion planning research, offering the first large-scale real-world dataset and evaluation schemes requiring both precise short-term planning and long-horizon ego-forecasting. Existing systems struggle to simultaneously meet both requirements. Indeed, we find that these tasks are fundamentally misaligned and should be addressed independently. We further assess the current state of closed-loop planning in the field, revealing the limitations of learning-based methods in complex real-world scenarios and the value of simple rule-based priors such as centerline selection through lane graph search algorithms. More surprisingly, for the open-loop sub-task, we observe that the best results are achieved when using only this centerline as scene context (i.e., ignoring all information regarding the map and other agents). Combining these insights, we propose an extremely simple and efficient planner which outperforms an extensive set of competitors, winning the nuPlan planning challenge 2023.

**Keywords:** Motion Planning, Autonomous Driving, Data-driven Simulation

## 1 Introduction

Despite learning-based systems' success in vehicle motion planning research [1, 2, 3, 4, 5], a lack of standardized large-scale datasets for benchmarking holds back their transfer from research to applications [6, 7, 8]. The recent release of the nuPlan dataset and simulator [9], a collection of 1300 hours of real-world vehicle motion data, has changed this, enabling the development of a new generation of learned motion planners, which promise reduced manual design effort and improved scalability. Equipped with this new benchmark, we perform the first rigorous empirical analysis on a large-scale, open-source, and data-driven simulator for vehicle motion planning, including a comprehensive set of state-of-the-art (SoTA) planners [10, 11, 12] using the official metrics. Our analysis yields several surprising findings:

**Open- and closed-loop evaluation are misaligned.** Most learned planners are trained through the supervised learning task of forecasting the ego vehicle's future motion conditioned on a desired goal location. We refer to this setting as ego-forecasting [2, 3, 13, 14]. In nuPlan, planners can be evaluated in two ways: (1) in open-loop evaluation, which measures ego-forecasting accuracy using distance-based metrics or (2) in closed-loop evaluation, which assesses the actual driving performance in simulation with metrics such as progress or collision rates. Open-loop evaluation lacks dynamic feedback and can have little correlation with closed-loop driving, as previously shown on the simplistic CARLA simulator [15, 16]. Our primary contribution lies in uncovering a *negative correlation* between both evaluation schemes. Learned planners excel at ego-forecasting but struggle to make safe closed-loop plans, whereas rule-based planners exhibit the opposite trend.

**Rule-based planning generalizes.** We surprisingly find that an established rule-based planning baseline from over twenty years ago [17] surpasses *all SoTA learning-based methods* in terms of closed-loop evaluation metrics on our benchmark. This contradicts the prevalent motivating claim used in most research on learned planners that rule-based planning faces difficulties in generalization.

7th Conference on Robot Learning (CoRL 2023), Atlanta, USA.

This was previously only verified on simpler benchmarks [4, 10, 11]. As a result, most current work on learned planning only compares to other learned methods, ignoring rule-based baselines [3, 5, 18].

**A centerline is all you need for ego-forecasting.** We implement a naïve learned planning baseline which does not incorporate any input about other agents in the scene and merely extrapolates the ego state given a centerline representation of the desired route. This baseline *sets the new SoTA* for open-loop evaluation on our benchmark. It does not require intricate scene representations (e.g. lane graphs, vectorized maps, rasterized maps, tokenized objects), which have been the central subject of inquiry in previous work [10, 11, 12]. None of these prior studies considered a simple centerline-only representation as a baseline, perhaps due to its extraordinary simplicity.

Our contributions are as follows: (1) We demonstrate and analyze the misalignment between open- and closed-loop evaluation schemes in planning. (2) We propose a lightweight extension of IDM [17] with real-time capability that achieves state-of-the-art closed-loop performance. (3) We conduct experiments with an open-loop planner, which is only conditioned on the current dynamic state and a centerline, showing that it outperforms sophisticated models with complex input representations. (4) By combining both models into a hybrid planner, we establish a simple baseline that outperformed 24 other, often learning-based, competing approaches and claimed victory in the nuPlan challenge 2023.

## 2 Related Work

**Rule-based planning.** Rule-based planners offer a structured, interpretable decision-making framework [17, 19, 20, 21, 22, 23, 24, 25, 26]. They employ explicit rules to determine an autonomous vehicle's behavior (e.g., brake when an object is straight ahead). A seminal approach in rule-based planning is the Intelligent Driver Model (IDM [17]), which is designed to follow a leading vehicle in traffic while maintaining a safe distance. There exist extensions of IDM [27] which focus on enabling lane changes on highways. However, this is not the goal of our work. Instead, we extend IDM by executing multiple policies with different hyperparameters, and scoring them to select the best option.

Prior work also combines rule-based decision-making with learned components, e.g., with learned agent forecasts [28], affordance indicators [23, 24], cost-based imitation learning [4, 29, 30, 31, 32], or learning-based planning with rule-based safety filtering [33]. These hybrid planners often forecast future environmental states, enabling informed and contingent driving decisions. This forecasting can either be agent-centric [34, 35, 36], where trajectories are determined for each actor, or environment-centric [4, 31, 30, 29, 37, 38], involving occupancy or cost maps. Additionally, forecasting can be conditioned on the ego-plan, modeling the ego vehicle's influence on the scene's future [39, 40, 41, 42]. We employ an agent-centric forecasting module that is considerably simpler than existing methods, allowing for its use as a starting point in the newly released nuPlan framework.

**Ego-forecasting.** Unlike predictive planning, ego-forecasting methods use observational data to directly determine the future trajectory. Ego-forecasting approaches include both end-to-end methods [43] that utilize LiDAR scans [44, 45], RGB images [46, 47, 48, 49, 14, 50] or both [13, 51, 5, 52], as well as modular methods involving lower-dimensional inputs like bird's eye view (BEV) grids or state vectors [24, 53, 11, 54, 55, 56]. A concurrent study introduces a naive MLP inputting the current dynamic state, yielding competitive ego-forecasting results on the nuScenes dataset [57] with no scene context input [58]. Our findings complement these results, differing by evaluating long-term (8s) ego-forecasting in the challenging 2023 nuPlan challenge scenario test distribution [9]. We show that in this setting, completely removing scene context (as in [58]) is harmful, whereas a simple centerline representation of the context is sufficient for strong open-loop performance.

## 3 Ego-forecasting and Planning are Misaligned

In this section, we provide the relevant background regarding the data-driven simulator nuPlan [9]. We describe two baselines for a preliminary experiment to demonstrate that although ego-forecasting and planning are often considered related tasks, they are not well-aligned given their definitions on nuPlan. Improvements in one task can often lead to degradation in the other.

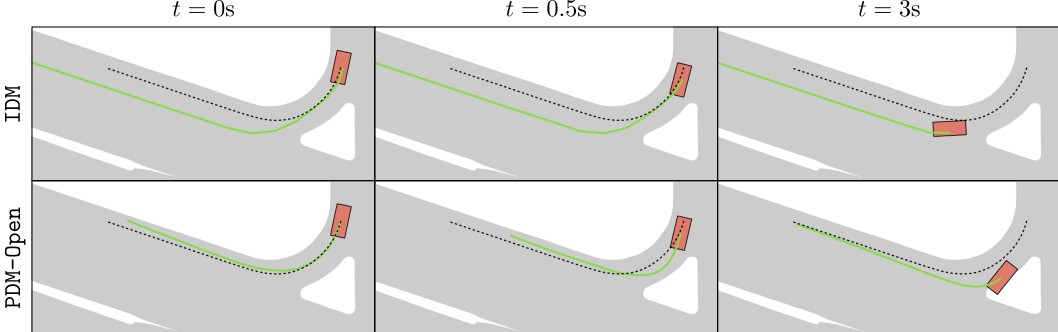

Figure 1: **Planning vs. ego-forecasting.** We present a nuPlan scenario, highlighting the driveable area in grey and the original human trajectory as a dashed black line. In each snapshot, we display the ego agent with its prediction. (Left) observe the significant displacement between the IDM prediction (constrained to a rule-based centerline) and the human trajectory, resulting in low open-loop scores. (Mid + right) after 0.5 seconds of simulation, the learned PDM-Open planner extrapolates its own errors and eventually veers off-road, leading to suboptimal closed-loop scores.

## 3.1 Background

**nuPlan.** The nuPlan simulator is the first publicly available real-world planning benchmark and enables rapid prototyping and testing of motion planners. nuPlan constructs a simulated environment as closely as possible a real-world driving setting through data-driven simulation [59, 60, 61, 62, 63, 64, 65]. This method extracts road maps, traffic patterns, and object properties (positions, orientations, and speeds) from a pre-recorded dataset consisting of 1,300 hours of real-world driving. These elements are then used to initialize scenarios, which are 15-second simulations employed to assess open-loop and closed-loop driving performance. Hence, in simulation, our methods rely on access to detailed HD map information and ground-truth perception. i.e., no localization errors, map imperfections, or misdetections are considered. In open-loop simulation, the entire log is merely replayed (for both the ego vehicle and other actors). Conversely, in closed-loop simulation, the ego vehicle operates under the control of the planner being tested. There are two versions of closed-loop simulation: non-reactive, where all other actors are replayed along their original trajectory, and reactive, where other vehicles employ an IDM planner [17], which detail in the following.

**Metrics.** nuPlan offers three official evaluation metrics: open-loop score (OLS), closed-loop score non-reactive (CLS-NR), and closed-loop score reactive (CLS-R). Although CLS-NR and CLS-R are computed identically, they differ in background traffic behavior. Each score is a weighted average of sub-scores that are multiplied by a set of penalties. In OLS, the sub-scores account for displacement and heading errors, both average and final, over an extended period (8 seconds). Moreover, if the prediction error is above a threshold, the penalty results in an OLS score of zero for that scenario. Similarly, sub-scores in CLS comprise time-to-collision, progress along the experts' route, speed-limit compliance, and comfort. Multiplicative CLS penalties are at-fault collisions, driveable area or driving direction infringements and not making progress. These penalties result in substantial CLS reductions, mostly to a zero scenario score, e.g., when colliding with a vehicle. Notably, the CLS primarily relies on short-term actions rather than on consistent long-term planning. All scores (incl. OLS/CLS) lie range from 0 to 100, where higher scores are better. Given the elaborate composition of nuPlan's metrics, we refer to the supplementary material for a detailed description.

**Intelligent Driver Model.** The simple planning baseline IDM [17] not only simulates the non-ego vehicles in the CLS-R evaluation of nuPlan, but also serves as a baseline for the ego-vehicle's planning. The nuPlan map is provided as a graph, with centerline segments functioning as nodes. After choosing a set of such nodes to follow via a graph search algorithm, IDM infers a longitudinal trajectory along the selected centerline. Given the current longitudinal position $x$, velocity $v$, and distance to the leading vehicle $s$ along the centerline, it iteratively applies the following policy to

calculate a longitudinal acceleration:

$$\frac{\mathrm{d}v}{\mathrm{d}t} = a\Big(1 - \Big(\frac{v}{v_0}\Big)^{\delta} - \Big(\frac{s^*}{s}\Big)^{2}\Big). \tag{1}$$

The acceleration limit $a$, target speed $v_0$, safety margin $s^*$, and exponent $\delta$ are manually selected. Intuitively, the policy uses an acceleration $a$ unless the velocity is already close to $v_0$ or the leading vehicle is at a distance of only $s^*$. Additional details and our exact hyper-parameter choices can be found in the supplementary material.

## 3.2  Misalignment

**Centerline-conditioned ego-forecasting.**  We now propose the Predictive Driver Model (Open), i.e., PDM-Open, which is a straightforward multi-layer perceptron (MLP) designed to predict future waypoints. The inputs to this MLP are the centerline ($\mathbf{c}$) extracted by IDM and the ego history ($\mathbf{h}$). To accommodate the high speeds (reaching up to 15 m/s) and ego-forecasting horizons (extending to 8 seconds) observed in nuPlan, the centerline is sampled with a resolution of 1 meter up to a length of 120 meters. Meanwhile, the ego history incorporates the positions, velocities, and accelerations of the vehicle over the previous two seconds, sampled at a rate of 5Hz. Both $\mathbf{c}$ and $\mathbf{h}$ are linearly projected to feature vectors of size 512, concatenated, and input to the MLP $\phi_{\texttt{Open}}$ which has two 512-dimensional hidden layers. The output are the future waypoints for an 8-second horizon, spaced 0.5 seconds apart, expressed as $\mathbf{w}_{\texttt{Open}} = \phi_{\texttt{Open}}(\mathbf{c}, \mathbf{h})$. The model is trained using an $L_1$ loss on our training dataset of 177k samples (described in Section 4). By design, PDM-Open is considerably simpler than existing learned planners [10, 12].

**OLS vs. CLS.**  In Table 1, we benchmark the IDM and PDM-Open baselines using the nuPlan metrics. We present two IDM variants with different maximum acceleration values (the default $a = 1.0\text{ms}^{-2}$ and $a = 0.1\text{ms}^{-2}$) and four PDM-Open variants based on different inputs. We observe that reducing IDM's acceleration improves OLS but negatively impacts CLS. While IDM demonstrates strong closed-loop performance, PDM-Open outperforms IDM in open-loop even if it only uses the current ego state as input (first row). The past ego states (History) only yield little improvement and lead to a drop in CLS. Most importantly, adding the centerline significantly contributes to ego-forecasting performance. A clear trade-off between CLS and OLS indicates a misalignment between the goals of ego-forecasting and planning. This sort of inverse correlation on nuPlan is unanticipated, considering the increasing use of ego-forecasting in current planning literature [3, 10, 12, 11]. While ego-forecasting is not necessary for driving performance, the nuPlan challenge requires both a high OLS and CLS.

| Method | Centerline | History | CLS-R↑ | CLS-NR↑ | OLS↑ |
|---|---|---|---|---|---|
| IDM [17] $a = 1.0$ | ✓ | - | **77** | **76** | 38 |
| $a = 0.1$ | | | 54 | 66 | 48 |
| PDM-Open | - | - | 51 | 50 | 69 |
| | - | ✓ | 38 | 34 | 72 |
| | ✓ | - | 54 | 53 | 85 |
| | ✓ | ✓ | 54 | 50 | **86** |

Table 1: **OLS-CLS Tradeoff.** Baseline scores on nuPlan with different inputs.

In Fig. 1, we illustrate the misalignment between the OLS and CLS metrics. In the depicted scenario, the rule-based IDM selects a different lane in comparison to the human driver. However, it maintains its position on the road throughout the simulation. This results in a high CLS yet a low OLS. Conversely, the learned PDM-Open generates predictions along the lane chosen by the human driver, thereby obtaining a high OLS. Nonetheless, as errors accumulate in its short-term predictions during the simulation [66, 67], the model's trajectory veers off the drivable area, culminating in a subpar CLS.

## 3.3  Methods

We now extend IDM by incorporating several concepts from model predictive control, including forecasting, proposals, simulation, scoring, and selection, as illustrated in Fig. 2 (top). We call this model PDM-Closed. Note that as a first step, we still require a graph search to find a sequence of lanes along the route and extract their centerline, as in the IDM planner.

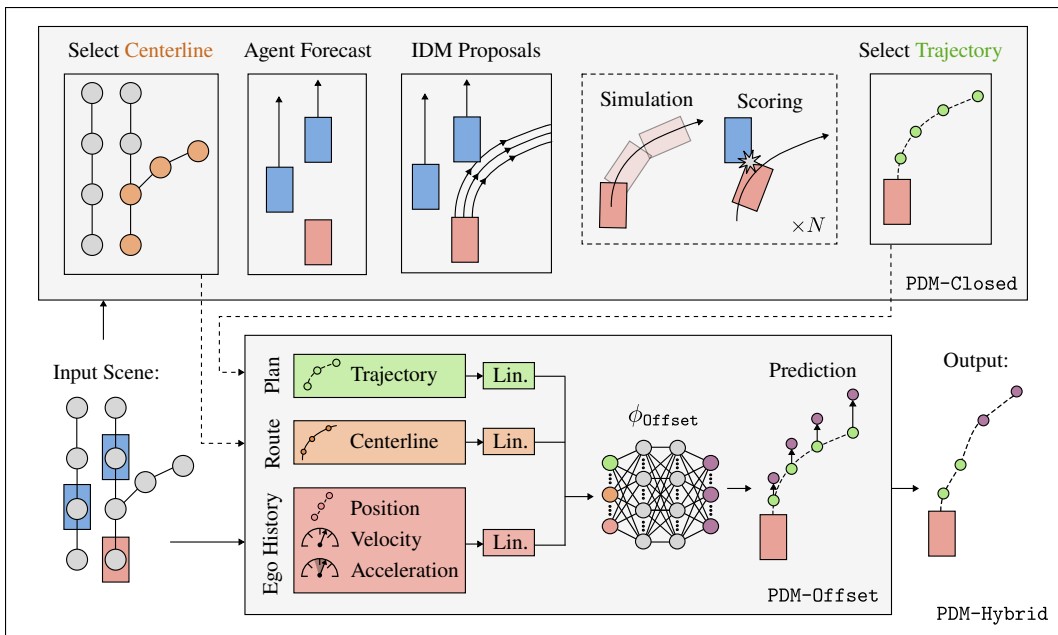

Figure 2: **Architecture.** PDM-Closed selects a centerline, forecasts the environment, and creates varying trajectory proposals, which are simulated and scored for trajectory selection. The PDM-Hybrid module predicts offsets using the PDM-Closed centerline, trajectory, and ego history, correcting only long-term waypoints and thereby limiting the learned model's influence in closed-loop simulation.

**Forecasting.** In nuPlan, the simulator provides an orientation vector and speed for each dynamic agent such as a vehicle or pedestrian. We leverage a simple yet effective constant velocity forecasting [68, 69, 70] over the horizon $F$ of 8 seconds at 10Hz.

**Proposals.** In the process of calibrating the IDM planner, we observed a trade-off when selecting a single value for the target speed hyperparameter ($v_0$), which either yielded aggressive driving behavior or insufficient progress across various scenarios. Consequently, we generate a set of trajectory proposals by implementing IDM policies at five distinct target speeds, namely, $\{20\%, 40\%, 60\%, 80\%, 100\%\}$ of the designated speed limit. For each target speed, we also incorporate proposals with three lateral centerline offsets ($\pm 1$m and 0m), thereby producing $N = 15$ proposals in total. To circumvent computational demands in subsequent stages, the proposals have a reduced horizon of $H$ steps, which corresponds to 4 seconds at a 10Hz.

**Simulation.** Trajectories in nuPlan are simulated by iteratively retrieving actions from an LQR controller [71] and propagating the ego vehicle with a kinematic bicycle model [72, 73]. We simulate the proposals with the same parameters and a faster re-implementation of this two-stage pipeline. Thereby, the proposals are evaluated based on the expected movement in closed-loop.

**Scoring.** Each simulated proposal is scored to favor traffic-rule compliance, progress, and comfort. By considering proposals with lateral and longitudinal variety, the planner can avoid collisions with agent forecasts and correct drift that may arise when the controller fails to accurately track the intended trajectory. Furthermore, our scoring function closely resembles the nuPlan evaluation metrics. We direct the reader to the supplementary material for additional details.

**Trajectory selection.** Finally, PDM-Closed selects the highest-scoring proposal which is extended to the complete forecasting horizon $F$ with the corresponding IDM policy. If the best trajectory is expected to collide within 2 seconds, the output is overwritten with an emergency brake maneuver.

**Enhancing long-horizon accuracy.** To integrate the accurate ego-forecasting capabilities of PDM-Open with the precise short-term actions of PDM-Closed, we now propose a hybrid version of PDM, i.e., PDM-Hybrid. Specifically, PDM-Hybrid uses a learned module PDM-Offset to predict offsets to waypoints from PDM-Closed, as shown in Fig. 2 (bottom).

| Method | Rep. | CLS-R ↑ | CLS-NR ↑ | OLS ↑ | Time ↓ |
|---|---|---|---|---|---|
| Urban Driver [10] | Polygon | 50 | 53 | 82 | 64 |
| GC-PGP [12] | Graph | 55 | 59 | 83 | 100 |
| PlanCNN [11] | Raster | 72 | 73 | 64 | 43 |
| IDM [17] | Centerline | 77 | 76 | 38 | 27 |
| PDM-Open | Centerline | 54 | 50 | **86** | **7** |
| PDM-Closed | Centerline | **92** | **93** | 42 | 91 |
| PDM-Hybrid | Centerline | **92** | **93** | 84 | 96 |
| PDM-Hybrid* | Graph | **92** | **93** | 84 | 172 |
| *Log Replay* | *GT* | *80* | *94* | *100* | *-* |

Table 2: **Val14 benchmark.** We show the closed-loop score reactive/non-reactive (CLS-R/CLS-NR), open loop score (OLS) and runtime in ms for several planners. We specify the input representation (Rep.) used by each planner. PDM-Hybrid accomplishes strong ego-forecasting (OLS) and planning (CLS). *This is a preliminary version of PDM-Hybrid that combined PDM-Closed with GC-PGP [12], and was used in our online leaderboard submission (Table 3).

In practice, the LQR controller used in nuPlan relies exclusively on the first 2 seconds of the trajectory when determining actions in closed-loop. Therefore, applying the correction only to long-term waypoints (i.e., beyond 2 seconds by default, which we refer to as the correction horizon $C$) allows PDM-Hybrid to maintain closed-loop planning performance. The final planner outputs waypoints (up to the forecasting horizon $F$) $\{\mathbf{w}_{\texttt{Hybrid}}^t\}_{t=0}^{F}$ that are given by:

$$\mathbf{w}_{\texttt{Hybrid}}^t = \mathbf{w}_{\texttt{Closed}}^t + \mathbb{1}_{[t>C]}\phi_{\texttt{Offset}}^t(\mathbf{w}_{\texttt{Closed}}, \mathbf{c}, \mathbf{h}). \quad (2)$$

Where $\mathbf{c}$ and $\mathbf{h}$ are the centerline and history (identical to the inputs of PDM-Open). $\{\mathbf{w}_{\texttt{Closed}}^t\}_{t=0}^{F}$ are the PDM-Closed waypoints added to the hybrid approach, and $\phi_{\texttt{Offset}}$ is an MLP. Its architecture is identical to $\phi_{\texttt{Open}}$ except for an extra linear projection to accommodate $\mathbf{w}_{\texttt{Closed}}$ as an additional input.

It is important to note that PDM-Hybrid is designed with high modularity, enabling the substitution of individual components with alternative options when diverse requirements emerge. For example, we show results with a different open-loop module in the supplementary material. Given its overall simplicity, one interesting approach to explore involves incorporating modular yet differentiable algorithms as components, as seen in [34]. Exploring the integration of these modules within unified multi-task architectures is another interesting direction. We reserve such exploration for future work.

## 4 Experiments

We now outline our proposed benchmark and highlight the driving performance of our approach.

**Val14 benchmark.** We offer standardized data splits for training and evaluation. Training uses all 70 scenario types from nuPlan, restricted to a maximum of 4k scenarios per type, resulting in ~177k training scenarios. For evaluation, we use 100 scenarios of the 14 scenario types considered by the leaderboard, totaling 1,118 scenarios. Despite minor imbalance (all 14 types do not have 100 available scenarios), our validation split aligns with the online leaderboard evaluation (Table 2 and Table 3), confirming the suitability of our Val14 benchmark as a proxy for the online test set.

**Baselines.** We include several additional SoTA approaches adopting ego-forecasting for planning in our study. Urban Driver [10] encodes polygons with PointNet layers and predicts trajectories with a linear layer after a multi-head attention block. Our study uses an implementation of Urban Driver trained in the open-loop setting. GC-PGP [12] clusters trajectory proposals based on route-constrained lane-graph traversals before returning the most likely cluster center. PlanCNN [11] predicts waypoints using a CNN from rasterized grid features without an ego state input. It shares several similarities to ChauffeurNet [8], a seminal work in the field. A preliminary version of PDM-Hybrid, which won the nuPlan competition, used GC-PGP as its ego-forecasting component, and we include this as a baseline. We provide a complete description of this version in the supplementary.

**Results.** Our results are presented in Table 2. `PlanCNN` achieves the best CLS among learned planners, possibly due to its design choice of removing ego state from input, trading OLS for enhanced CLS. Contrary to the community's growing preference for graph- and vector-based scene representations in prediction and planning [74, 11, 75, 76], these results show no clear disadvantage of raster representations for the closed-loop task, with `PlanCNN` also offering a lower runtime. Surprisingly, the simplest rule-based approach in our study, `IDM`, outperforms the best learned planner, `PlanCNN`. Moreover, we observe `PDM-Closed`'s advantages over `IDM` in terms of CLS: an improvement from 76-77 to 92-93 as a result of the ideas from Section 3. Surprisingly, `PDM-Open` achieves the highest OLS of 86 with a runtime of only 7ms using only a centerline and the ego state as input. We observe that `PDM-Open` improves on other methods in accurate long-horizon lane-following, as detailed further in our supplementary material. Next, despite `PDM-Closed`'s unsatisfactory 42 OLS, `PDM-Hybrid` successfully combines `PDM-Closed` with `PDM-Open`. Both the centerline and graph versions of `PDM-Hybrid` achieve identical scores in our evaluation. However, the final centerline version, using `PDM-Open` instead of `GC-PGP`, is more efficient during inference. Finally, the privileged approach of outputting the ground-truth ego future trajectory (log replay) fails to achieve a perfect CLS, in part due to the nuPlan framework's LQR controller occasionally drifting from the provided trajectory. `PDM-Hybrid` compensates for this by evaluating proposals based on the expected controller outcome, causing it to match/outperform log replay in closed-loop evaluation.

**Challenge.** The 2023 nuPlan challenge saw the preliminary (graph) version of `PDM-Hybrid` rank first out of 25 participating teams. The leaderboard considers the mean of CLS-R, CLS-NR, and OLS. While open-loop performance lagged slightly, closed-loop performance excelled, resulting in an overall SoTA score. Unfortunately, due to the closure of the leaderboard, our final (centerline) version of `PDM-Hybrid` that replaces `GC-PGP` with the simpler `PDM-Open` module could not be benchmarked. All top contenders combined learned ego-forecasting with rule-based post-solvers or post-processing to boost CLS performance for the challenge [77, 78, 79]. Thus, we expect to see more hybrid approaches in the future.

| Method | CLS-R ↑ | CLS-NR ↑ | OLS ↑ | Score ↑ |
|---|---|---|---|---|
| PDM-Hybrid* | **93** | **93** | 83 | **90** |
| hoplan | 89 | 88 | 85 | 87 |
| pegasus_multi_path | 82 | 85 | **88** | 85 |
| Urban Driver [10] | 68 | 70 | 86 | 75 |
| IDM [17] | 72 | 75 | 29 | 59 |

Table 3: **2023 nuPlan Challenge.**

Importantly, near identical scores were recorded for our submission on both our Val14 benchmark (Table 2) and the official leaderboard (Table 3). Note that the `Urban Driver` and `IDM` results on the leaderboard are provided by the nuPlan team, so they likely use different training data and hyper-parameters than our implementations from Table 2.

**Ablation Study.** We delve into our design choices through an ablation study in Table 4. Table 4a displays `PDM-Hybrid`'s closed-loop score reactive (CLS-R) and open-loop score (OLS) with varied correction horizons ($C$) from 0s to 3s. Applying the waypoint correction to all waypoints (i.e., $C = 0$), outperforms `PDM-Open` in OLS (87 vs. 86, see Table 2) but leads to a substantial drop in CLS-R compared to the default value of $C = 2$. On the other hand, a noticeable OLS decline occurs when initiating corrections deeper into the trajectory (e.g., $C = 3$), with minimal impact on CLS-R.

For `PDM-Closed` (Table 4b), we compare CLS-R and runtime (ms) with the base planner across three scenarios: removing lateral centerline offsets ("lat."), longitudinal IDM proposals ("lon."), and environment forecasting ("cast."). Our analysis reveals that eliminating proposals diminishes CLS-R effectiveness but accelerates runtimes. Performance significantly drops when excluding the forecasting used for creating and evaluating proposals. However, the runtime remains nearly identical, showing the effectiveness of the simple forecasting mechanism.

As for `PDM-Open` (Table 4c), we test three variations: a shorter centerline (30m vs. 120m), a coarser centerline (every 10m vs. 1m), and a smaller MLP with a reduced hidden dimension (from 512 to 256). Both a smaller MLP and a reduced centerline length lead to performance degradation, but the impact remains relatively minor compared to disregarding the centerline altogether (Table 1, OLS=72). Meanwhile, the impact of a coarser centerline is negligible.

| $C$ | CLS-R ↑ | OLS ↑ |
|---|---|---|
| 0.0s | 58 | **87** |
| 2.0s | **92** | 84 |
| 2.5s | **92** | 84 |
| 3.0s | **92** | 72 |

(a) `PDM-Hybrid`

| Method | CLS-R ↑ | Time ↓ |
|---|---|---|
| Base | **92** | 91 |
| No lat. | 89 | **55** |
| No lon. | 88 | 64 |
| No cast. | 86 | 90 |

(b) `PDM-Closed`

| Method | OLS ↑ |
|---|---|
| Baseline | **86** |
| Shorter centerline | 84 |
| Coarser centerline | **86** |
| Smaller MLP | 84 |

(c) `PDM-Open`

Table 4: **Ablation Study.** We show the closed-loop score reactive (CLS-R), open loop score (OLS) and runtime in ms. We investigate (a) varying correction horizons for `PDM-Hybrid`, (b) ignoring sub-modules of `PDM-Closed`, and (c) the effects of input and architecture choices on `PDM-Open`. The default configuration, highlighted in gray, achieves the best trade-offs.

## 5 Discussion

Although rule-based planning is often criticized for its limited generalization, our results demonstrate strong performance in the closed-loop nuPlan task which best resembles real-world evaluation. Notably, open-loop success in part requires a *trade-off in closed-loop performance*. Consequently, imitation-trained ego-forecasting methods fare poorly in closed-loop. This suggests that rule-based planners remain promising and warrant further exploration. At the same time, given their poor performance out-of-the-box, there is room for improvement in imitation-based methods on nuPlan.

Integrating the strengths of closed-loop planning and open-loop ego-forecasting, we present a hybrid model. However, this does not enhance closed-loop driving performance; instead, it boosts open-loop performance *while executing identical driving maneuvers*. We conclude that considering precise open-loop ego-forecasting as a prerequisite for achieving long-term planning goals is misleading.

Acknowledging the potential importance of ego-forecasting for interpretability and assessing human-like behavior, we propose focusing this evaluation on the short horizon (e.g., 2 seconds) relevant for closed-loop driving. The current nuPlan OLS definition, requiring a unimodal 8-second ego-forecast, may only be useful for *alternate applications*, like setting goals for background agents in data-driven traffic simulations or allocating computational resources better, e.g. to prioritize perception or prediction in areas the ego-vehicle is expected to traverse. We discourage the use of open-loop metrics as a primary indicator of planning performance [18].

**Limitations.** While we significantly improve upon the established `IDM` model, PDM still does not execute lane-change maneuvers. Lane change attempts often lead to collisions when the ego-vehicle is between two lanes, resulting in a high penalty as per the nuPlan metrics. PDM relies on HD maps and precise offboard perception [80, 9] that may be unavailable in real-world driving situations. While real-world deployment was demonstrated for learning-based methods [81, 33, 8], it remains a significant challenge for rule-based approaches. Moreover, our experiments, aside from the held-out test set, have not specifically evaluated the model's generalization capabilities when encountering distributional shifts, such as unseen towns or novel scenario types. They were all conducted on a single simulator, nuPlan. Therefore, it is important to recognize the limitations inherent in nuPlan's data-driven simulation approach. When a planner advances more rapidly than the human driving log, objects materialize abruptly in front of the ego-vehicle during simulation. For CLS-NR, vehicles move independently as observed in reality, disregarding the ego agent, leading to excessively aggressive behavior. Conversely, CLS-R background agents rely on `IDM` and adhere strictly to the centerline, leading to unrealistically passive behavior. We see high value in developing a more refined reactive environment for future work.

**Conclusion.** In this paper, we identify prevalent misconceptions in learning-based vehicle motion planning. Based on our insights, we introduce `PDM-Hybrid`, which builds upon `IDM` and combines it with a learned ego-forecasting component. It surpassed a comprehensive set of competitors and claimed victory in the 2023 nuPlan competition.

**Acknowledgements**

Andreas Geiger was supported by the ERC Starting Grant LEGO-3D (850533), the BMWi in the project KI Delta Learning (project number 19A19013O) and the DFG EXC number 2064/1 - project number 390727645. Kashyap Chitta was supported by the German Federal Ministry of Education and Research (BMBF): Tübingen AI Center, FKZ: 01IS18039A. We thank the International Max Planck Research School for Intelligent Systems (IMPRS-IS) for supporting Kashyap Chitta. We also thank Pat Karnchanachari and Gianmarco Bernasconi for helping with technical issues in our leaderboard submissions, Niklas Hanselmann for proofreading, and Andreas Zell for helpful discussions.

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
