# OpenReview forum: "Parting with Misconceptions about Learning-based Vehicle Motion Planning"
_robot-learning.org/CoRL/2023/Conference — CoRL 2023 Poster_

### Official Review · Reviewer_qggz · 2023-07-19

**Confidence:** 4
**Originality:** Fair
**Technical Quality:** Good
**Clarity Of Presentation:** Good
**Impact:** 3

**Recommendation:**

Weak Reject: I recommend rejecting the paper, but will not argue for my recommendation if the majority of other reviewers have a different opinion.

**Review:**

The paper is well written and easy to follow. The paper's core contributions around revealing evaluation issues and limitations of learning methods are quite original. However, some ideas like using rule-based planners as a baseline have been explored before. The proposed PDM-Hybrid model is simple and high-performing but not an entirely new technique. The experimental results are comprehensive but significance is somewhat limited by focusing only on one simulator (nuPlan) and competition-style metrics. The findings may not generalize fully to other environments and tasks.

The revealing analysis and actionable recommendations do provide valuable new insights that stand to advance autonomous driving research. However, the proposed techniques themselves represent only an incremental advance, as evidenced by the simple model achieving state-of-the-art results. Substantial innovation of new planning and learning techniques is still needed for broader progress.

Strengths:
+ The analysis in the paper reveals fundamental misalignments between open-loop ego-forecasting and closed-loop planning assumptions. It also shows limitations of learning methods compared to a simple rule-based approach.
+ The introduced PDM-Hybrid model achieves state-of-the-art results on nuPlan, winning the 2023 challenge. Its modular design is also flexible.
+ The paper makes specific recommendations for future research, such as focusing ego-forecasting evaluation on short 2 second horizons relevant to control, rather than unrealistic long 8 second predictions.

Weakness:
+ The paper lacks clarity regarding its contributions, making it difficult to discern the originality of the work and distinguish between incremental and novel aspects. The absence of a conclusion section further compounds this issue. It is recommended that the authors include a dedicated paragraph outlining their contributions and incorporate a conclusion section to address these concerns.
+ The modular PDM-Hybrid architecture exhibits limitations, particularly in the fact that its components are not learned end-to-end, potentially hindering the full utilization of crosstalk between different submodules.
+ The absence of information regarding the computational platform used for the simulations is a notable omission. Moreover, the reliance on access to HD maps and perfect perception raises questions about the current real-world applicability of the proposed method. Addressing imperfect state estimation and maps is crucial. Therefore, the authors should include discussions on how their proposed method could be extended to real-world scenarios. Additionally, conducting further evaluations in diverse and interactive simulators would enhance the assessment of generalizability.


**Quality Of The Limitations Section:**

Limitations are addressed clearly

**Questions For Rebuttal:**

+ Please clearly highlight the novelty of the proposed approach, what is the contributions, and what is being distinguish between incremental and novel aspects. Please also provide concluding remarks on this paper.
+ Explicitly state the assumptions made in the problem formulation.
+ The proposed PDM-Hybrid model seems incremental - how is it substantially different from prior techniques? What innovations does it introduce?
+ Please comment on the computations cost of the proposed approach and on what computational platform was this approach implemented.
+ How do you ensure safety guarantees for the learned components of your model?
+ Does your approach account for imperfect state estimation or map errors that would occur in practice?

**Robotics Focus:**

Relevant but unlikely to deploy to hardware in near future

**Summary Of Paper:**

This research paper investigates the alignment between open-loop trajectory forecasting and closed-loop planning metrics on the nuPlan benchmark in the context of autonomous driving, with learning-based planners excelling at the former but a simple 20-year-old rule-based planner performing best for the latter. Notably, a centerline-only model surpasses methods employing elaborate scene representations, indicating limited long-horizon context value in open-loop forecasting. The paper proposes PDM-Hybrid, which combines rule-based and learned components, yielding state-of-the-art performance on both metrics with enhanced simplicity and efficiency. Extensive experiments demonstrate the effectiveness of this approach.  The paper makes key recommendations like focusing forecasting evaluation on short control-relevant horizons and avoiding reliance on open-loop metrics for planning assessment. Code, data and models are released to facilitate future autonomous driving research.

**Summary Of Recommendation:**

Overall, I'm positive about this work, but there are some limitations in terms of novelty, rigor, and technical advancements. The core revelation of misaligned metrics and learning method issues is an excellent contribution. However, the proposed PDM-Hybrid model only offers minor advances compared to established techniques. The experiments are fairly rigorous but lack extensive ablation studies and testing in diverse simulation environments to demonstrate generalizability. While the actionable recommendations for the field are impactful, the paper doesn't propose substantial new techniques to achieve these goals. Additionally, the lack of physical robot validation raises concerns about real-world applicability.

---

### Official Review · Reviewer_KEQU · 2023-07-20

**Confidence:** 4
**Originality:** Very Good
**Technical Quality:** Excellent
**Clarity Of Presentation:** Excellent
**Impact:** 4

**Recommendation:**

Strong Accept: I recommend accepting the paper and will argue for my recommendation even if other reviewers hold a different opinion.

**Review:**

Strengths

-The paper is well-written and easy to follow with nice explanations of the technical components and a helpful discussion of results.

-The empirical study is quite extensive, containing ablation studies and in-depth discussions of the performance of the approaches evaluated.

-The findings are quite intriguing, demonstrating the value of rule-based approaches for some types of driving tasks. Through its rigorous evaluation, the paper follows a few recent works demonstrating the value of well-understood, model-based approaches. See Scholler et al. What the constant velocity model can teach us about pedestrian motion prediction. RA-L 2020; Poddar et al. From Crowd Motion Prediction to Robot Navigation in Crowds. arXiv:2303.01424. 2023. I've felt like there is a point to be made about this, and the authors execute this quite well.

Weaknesses

-Section 3 could be organized a little better. I feel like the introductory headers in bold are helpful but a better organization through grouped subsections could help readability.

-The scenarios highlighted are quite interesting but they feature relatively structured motion. Are there any insights for more interactive scenarios like (unsignalized) intersections? How would the PDM-based approach perform in such settings?

-A discussion of how the insights extracted would likely port over (or not) to the real world would be interesting to add.

***POST REBUTTAL***

The rebuttal addressed my comments and the additional supplementary material strengthened the paper. I remain in favor of acceptance.

**Quality Of The Limitations Section:**

Limitations are addressed clearly

**Questions For Rebuttal:**

-Discuss the connection of the insights extracted in the study to the real-world and the possibility of future hardware deployment.

-Discuss scaling to more complex scenarios and behaviors (e.g., intersections, aggressive drivers etc).

**Robotics Focus:**

Highly relevant to robotics but no hardware experiments

**Summary Of Paper:**

This paper presents an empirical study on the nuPlan dataset and benchmark. Specifically, the authors compare a series of planners ---including a proposed one that won the nuPlan challenge--- with respect to their open-loop and closed-loop performance. Through extensive ablation studies, the paper discusses key takeaways and directions for future work. One interesting insight is the observation that rule-based baselines exhibit strong performance in closed-loop settings in the nuPlan challenge, suggesting that they should not be prematurely rejected as non-generalizable approaches for some driving tasks.

**Summary Of Recommendation:**

Overall, this is a well written paper with compelling findings and well presented insights. With the incorporation of a few points of discussion (see above), I am in favor for acceptance.

---

### Official Review · Reviewer_xfB2 · 2023-07-22

**Confidence:** 3
**Originality:** Fair
**Technical Quality:** Fair
**Clarity Of Presentation:** Good
**Impact:** 2

**Recommendation:**

Weak Reject: I recommend rejecting the paper, but will not argue for my recommendation if the majority of other reviewers have a different opinion.

**Review:**

Strengths
========
- The paper is clearly written and reads well.
- The related work provides a nice overview of rule-based planning and approaches performing ego-forecasting.
- The figures are well made and illustrate the task and its complexity well.
- One of the approaches presented in the paper won the nuPlan 2023 challenge.



Constructive Criticism
=================
- I am unfamiliar with the nuPlan challenge and do not know the level of complexity it poses. From the visualizations in the paper and the supplementary video, the tested environments look very simplistic, it appears that algorithms have perfect knowledge of the environment. Does this challenge contain a perception component?
- While the study won the inaugural nuPlan challenge, it would be useful to know how the proposed method compares to other methods in more diverse and complex real-world scenarios. A comparison to additional datasets besides nuPlan could enhance the generalizability and robustness of the findings.
- The discussion section reads as if there are only two classes of planning approaches for autonomous vehicles: rule-based methods and approaches based on imitation learning. I'm no expert in the field, but I assume there are approaches that combine interpretable (i.e. close to rule-based) decision-making approaches with data-driven environment models. It would be interesting if the paper could extend on this.
- The paper refers to citation 59 for information about the evaluation metric's composition. Given that the metric is of such central nature to the findings of this paper, I suggest to include a description of it in the main text.

**Quality Of The Limitations Section:**

Limitations are addressed clearly

**Questions For Rebuttal:**

How would the findings presented in this paper generalize to a more realistic driving scenario where perception is noisy and incomplete?

How does the reported metric relate to crash rates? How does this work compare to the state of the art in autonomous driving? Seeing that the best-recorded score in this paper is 93% (given perfect perception), I'm not sure if this correctly reflects SOTA. There are already companies that perform autonomous driving on real vehicles in real-world environments that obviously reach success rates substantially higher than 99%.



**Robotics Focus:**

Relevant but unlikely to deploy to hardware in near future

**Summary Of Paper:**

The paper presents an empirical analysis of the performance of vehicle motion planning approaches in the nuPlan simulator.
In doing so, the paper focuses on the difference in performance achieved between evaluating in a "closed-loop scheme" and an "open-loop scheme". While "open-loop" evaluation can be done on a static dataset by comparing generated actions of an algorithm with ground-truth labels, closed-loop evaluation requires an actual simulator where progress and collision rates are measured.

The paper shows that there is a negative correlation between the two evaluation schemes when tested in the nuPlan simulator. Furthermore, the paper states that a simple rule-based planning baseline surpasses all SOTA learning-based methods.

**Summary Of Recommendation:**

My recommendation is mainly based on the fact that this paper tackles the task of motion planning in a setting where perception is perfect. Tackling the motion planning task without considering the challenges of perception seems disconnected from the state of the art.

---

### Official Review · Reviewer_7G8b · 2023-07-25

**Confidence:** 3
**Originality:** Very Good
**Technical Quality:** Very Good
**Clarity Of Presentation:** Very Good
**Impact:** 4

**Recommendation:**

Strong Accept: I recommend accepting the paper and will argue for my recommendation even if other reviewers hold a different opinion.

**Review:**

I think this paper offers interesting insight into the problem of autonomous driving. Mostly it suggests to me that rule-based planners combined with simple learning methods might still have a lot to offer. That said I do have the following concerns about it.

First, a key design decision in PDM-Hybrid is that the planner only modifies trajectories suggested by PDM-Closed beyond the horizon of the simulator’s closed-loop controller (LQR). This is what allowed the team to bring their score on ego-forecasting back up without impacting their closed-loop performance at all. I think this is clever, but I hesitate to cast this as any sort of insight into the problem of autonomous driving beyond pointing to the fact that ego-forecasting is probably a bad metric to use for future competitions. That said I think the authors acknowledge this towards the end of the paper. They might simply want to emphasize it earlier as well instead of introducing both metrics as equals involving a tradeoff.

My second concern with the paper is that it provides limited insight into what leads to good performance in the ego-forecasting task. With PDM-Open, the limited input (ego history and centerline) dramatically reduces the ability of the planner to learn any sort of policy (in contrast to what planners like PlanCNN - a competitive benchmark - are forced to do by removing ego from their inputs). It is therefore not surprising to me that this leads to accumulating error and poor performance in the closed-loop setting. What *is* surprising to me is that this leads to better performance in the ego-forecasting setting, and I think that the paper doesn’t really provide any insight as to why that might be the case.

Overall the paper is well written and the experiment satisfactory. I also think that the methods proposed are simple enough that they are likely to be reproduced by the community.

**Quality Of The Limitations Section:**

Limitations are addressed clearly

**Questions For Rebuttal:**

For the results provided in Table 1, what is the input to PDM-Open for the version that has neither a centerline or history? Is it only the current state of ego?

**Robotics Focus:**

Highly relevant to robotics but no hardware experiments

**Summary Of Paper:**

This paper describes the approach taken by the 2023 nuPlan winning team. It tracks two metrics: ego-forecasting and closed-loop performance.

The work puts forward three general approaches. The first one (PDM-Open) takes as input the centerline predicted by a rule-based planner (IDM), as well as the history of ego, and outputs the predicted ego trajectory. The second one (PDM-Closed) generates a collection of candidate trajectories by using a fixed set of hyperparameters and IDM. Candidate trajectories are then simulated, scored, and the best one is chosen for execution. The final (winning) approach (PDM-Hybrid) uses PDM-Closed to generate a plan and adds it as an input to a model similar to PDM-Open.

PDM-Open excels at ego-forecasting but underperforms at closed-loop evaluation, PDM-Closed is the opposite, and PDM-Hybrid preserves the performance of PDM-Closed while recovering the performance of PDM-Open on ego-forecasting.

**Summary Of Recommendation:**

Overall I think the experiments are thorough and the results thought provoking. The main strengths of the paper are the simplicity of its approaches and its very strong empirical results. I do not think that the weaknesses in some of the analysis should detract from the paper’s overall strong contributions.

---

### Author Response · Authors · 2023-08-11
**General Comment to all Reviewers 1/2**

# General Response
We express our gratitude to all reviewers for dedicating their time and providing valuable feedback. We are pleased that the reviewers value the “interesting” [[7G8b](https://openreview.net/forum?id=o82EXEK5hu6&noteId=U4nqusSgbq)] and “well presented” [[KEQU](https://openreview.net/forum?id=o82EXEK5hu6&noteId=1-2ygssPtUA)] insights of our paper with “original” [qggz] contributions while highlighting the “simplicity of its approaches” [[7G8b](https://openreview.net/forum?id=o82EXEK5hu6&noteId=U4nqusSgbq)]. Moreover, we were pleased that the reviewers described our paper as “well” [[7G8b](https://openreview.net/forum?id=o82EXEK5hu6&noteId=U4nqusSgbq), [KEQU](https://openreview.net/forum?id=o82EXEK5hu6&noteId=1-2ygssPtUA), [qggz](https://openreview.net/forum?id=o82EXEK5hu6&noteId=ZAQ1VwHunE)] or “clearly” [[xfB2](https://openreview.net/forum?id=o82EXEK5hu6&noteId=U4nqusSgbq)] written with “rigorous” [[KEQU](https://openreview.net/forum?id=o82EXEK5hu6&noteId=1-2ygssPtUA)] evaluations and “thorough” [[7G8b](https://openreview.net/forum?id=o82EXEK5hu6&noteId=U4nqusSgbq)] experiments.

In the following, we comment on how our insights can be generalized to real-world applications, which was a general point raised by multiple reviewers. In particular, we focus on the assumption of perfect perception. We address all other individual comments and concerns of the reviewers separately below.

Moreover, we have updated our manuscript to improve the clarity and structure, taking into account the minor comments. All changes are highlighted in red to ease reviewing.

### Generalizing the insights to real-world applications (e.g. with imperfect perception/mapping)

* **It is a common approach in planning research to assume the availability of ground truth information about detected/tracked objects, localization, and mapping**  [1,2,3,4]. For instance, CAPO [5] (ICRA’22), DriveIRL [6] (ICRA’23), and UrbanDriver [7] (CoRL’21) rely on ground-truth trajectories of agents in their approach. Similarly, KING [8] (ECCV’22), ROACH [9] (ICCV’21), and PlanT [10] (CORL’22) present several experiments with ground truth information taken from the CARLA simulator [11].
Not introducing sub-optimal perception modules helps to better understand key factors contributing to planning performance. For instance, optimizing and ablating the model architecture or input representations may not generalize between two different perception modules. Hence, research with perfect perception provides valuable clues for the design of the perception system, e.g., which representations are most useful for planning.
We agree that the study of planning using full driving systems with imperfect perception is an important future direction. This type of evaluation requires simulating sensor data such as cameras and LiDAR, which is currently not possible with data-driven simulation frameworks like nuPlan.

* **nuPlan is the first and only publicly available real-world planning benchmark**. The nuPlan benchmark focuses on planning in a diverse set of interactive and challenging scenarios. The dataset is based on logs recorded in the real world with vehicle-mounted sensors. Hence, The ground-truth perception input is generated with an offline labeling system which has access to future information and is too resource intensive for real-time use in the vehicle.
The recorded scenarios include complex and interactive scenarios such as unprotected turns, traversing crowded pick-up drop-off areas, performing lane changes, and driving in congested traffic. As the surrounding agents are controlled by a reactive IDM policy, the planner is evaluated in a truly interactive setting. We thank the reviewers for pointing out that our supplementary video doesn’t reflect the dataset's complexity.
We will shortly upload additional videos showcasing the complexity and diversity of the nuPlan dataset.

* **Generalization from a planning dataset with ground-truth perception inputs to real-world traffic was successfully demonstrated in prior work**. We note that DriveIRL [6] (ICRA’23), UrbanDriver [7] (CoRL’21), and ChauffeurNet [12] (RSS’19) were able to do this, but all required legal approval and significant expenses for their real-world experiments. DriveIRL [6] (ICRA’23) was trained on nuPlan and then successfully deployed in real-world driving, showing the suitability of nuPlan for transfer to real scenarios. We were unable to reproduce this method in our study since the authors did not release their code or share it upon contact. UrbanDriver [7] (CoRL’21) has publicly released code and is one of the baselines in our study. Similarly, ChauffeurNet [12] (RSS’19) is similar in its architecture and training to PlanCNN, one of our baselines.
We stress that our adoption of the nuPlan benchmark promotes generalization to real world applications, as the dataset was recorded in the real-world and comprises an interactive simulation.

---

> ### Author Response · Authors · 2023-08-11
> **General Comment to all Reviewers 2/2**
>
> ### References
> [1] Jaime Fernández Fisac , Eli Bronstein, Elis Stefansson, Dorsa Sadigh, S. Shankar Sastry and Anca D. Dragan. “Hierarchical Game-Theoretic Planning for Autonomous Vehicles.” In ICRA, 2019.
>
> [2] Bingyu Zhou, Wilko Schwarting, Daniela Rus and Javier Alonso-Mora. “Joint Multi-Policy Behavior Estimation and Receding-Horizon Trajectory Planning for Automated Urban Driving.” In ICRA, 2018.
>
> [3] Edward Schmerling, Karen Leung, Wolf Vollprecht and Marco Pavone. “Multimodal Probabilistic Model-Based Planning for Human-Robot Interaction.” In ICRA, 2018.
>
> [4] Sadigh, S. Shankar Sastry, Sanjit A. Seshia and Anca D. Dragan. “Planning for Autonomous Cars that Leverage Effects on Human Actions.” In RSS, 2016.
>
> [5] Rowan McAllister, Blake Wulfe, Jean Mercat, Logan Ellis, Sergey Levine, and Adrien Gaidon. ”Control-Aware Prediction Objectives for Autonomous Driving” In ICRA, 2022.
>
> [6] Phan-Minh, T., Howington, F., Chu, T. S., Tomov, M. S., Beaudoin, R. E., Lee, S. U., ... & Wolff, E. M. “DriveIRL: Drive in Real Life with Inverse Reinforcement Learning.”, In ICRA, 2023
>
> [7] Scheel, O., Bergamini, L., Wolczyk, M., Osiński, B., & Ondruska, P.. “Urban driver: Learning to drive from real-world demonstrations using policy gradients.”, In CoRL, 2022
>
> [8] Niklas Hanselmann, Katrin Renz, Kashyap Chitta, Apratim Bhattacharyya and Andreas Geiger. “KING: Generating Safety-Critical Driving Scenarios for Robust Imitation via Kinematics Gradients.” In ECCV, 2022.
>
> [9] Zhejun Zhang, Alexander Liniger, Dengxin Dai, Fisher Yu and Luc Van Gool. “End-to-End Urban Driving by Imitating a Reinforcement Learning Coach.” In ICCV, 2021.
>
> [10] Renz, K., Chitta, K., Mercea, O. B., Koepke, A. S., Akata, Z., & Geiger, A. PlanT: “Explainable Planning Transformers via Object-Level Representations”, In CoRL, 2023
>
> [11] Dosovitskiy, A., Ros, G., Codevilla, F., Lopez, A., & Koltun, V. CARLA: "An open urban driving simulator", In CoRL, 2017
>
> [12] Bansal, M., Krizhevsky, A., & Ogale, A. “Chauffeurnet: Learning to drive by imitating the best and synthesizing the worst”, In RSS, 2019

---

### Decision · Program_Chairs · 2023-08-30

**Decision:**

Accept (Poster)

**Comment:**

This paper conducts an empirical analysis of vehicle motion planning approaches in the nuPlan simulator. It specifically examines the difference in performance between using a "closed-loop scheme" and an "open-loop scheme". While the reviewers have generally positive about this work, they did identify some writing-related issues that should be addressed in the final version.